# OpenReview forum: "MMAU: A Massive Multi-Task Audio Understanding and Reasoning Benchmark"
_ICLR.cc/2025/Conference — ICLR 2025 Spotlight_

### Official Review · Reviewer_2krx · 2024-10-20

**Soundness:** 3
**Presentation:** 3
**Contribution:** 2
**Rating:** 6
**Confidence:** 4

**Summary:**

The paper introduces a new benchmark for evaluating multimodal audio-language models across speech, sound, and music domains, with tasks requiring both information extraction and complex reasoning. The benchmark comprises 10,000 audio clips paired with annotated questions and answers, challenging models to demonstrate proficiency in 27 distinct skills. The authors provide a comprehensive analysis of 18 models, revealing current limitations in multimodal understanding, with the best-performing model achieving only 53% accuracy, highlighting significant room for improvement in this area.

**Strengths:**

1. MMAU is novel in its scope and comprehensiveness, covering a wide range of tasks that target specific skills in multimodal audio understanding and reasoning, addressing a critical gap in existing benchmarks.
2. The authors enlisted real-world human experts, conducted thorough evaluations, and used GPT-4 to enrich the dataset with additional options. Detailed prompt information is also provided in the appendix.

**Weaknesses:**

1. While the authors claim that tasks may require multiple skills, a review of Appendix H reveals that most tasks can be completed using a single skill. This claim lacks precision and is not sufficiently supported in the paper.
2. The data is sourced from 13 datasets, but the paper does not provide enough explanation regarding the selection of the 10,000 data points from these large datasets. In Section 3.2, the authors mention that experts were involved in classifying abilities and question-answer pairs, but there is no clear description of the selection process. How was the initial screening conducted, and how were quality and diversity ensured? Even Appendix E provides only a brief overview of the datasets, raising concerns about data quality.
3. Although expert annotation is mentioned, there are no specific details on how bias was controlled during the annotation process. The accuracy and consistency of benchmark data are central to evaluation reliability, but Appendix C.2 lacks elaboration on how expert annotations were quality-checked and bias-mitigated.
4. In the experimental section, while bias in model selection was avoided, the response quality of most models (e.g., SALAMONN and Qwen2-Audio) was significantly influenced by the prompts. These models were primarily trained in a QA format, and there is a noticeable difference between their open-ended responses and multiple-choice selections. This raises concerns about whether the evaluation method accurately reflects the models' limitations, and further discussion on this issue is necessary.
5. The evaluation in the experimental section relies on multiple-choice questions and uses string matching to determine correct answers. This method seems overly simplistic. Given the openness of LLM responses, a more sophisticated approach should be used to evaluate whether a model's output is sufficiently similar to the correct answer. The authors should provide more details about the evaluation methodology to ensure results are reliably reproducible.

**Questions:**

1. The paper mentions the distinction between reasoning and extraction tasks. However, some tasks appear to require a combination of both skills inherently. What is the rationale behind maintaining this distinction?
2. In Appendix L, the prompt requires the model to generate a correct option. What is the purpose of this approach, and how does it impact the evaluation?

---

> ### Author Response · Authors · 2024-11-20
> **Response to Official Review of Submission3927 by Reviewer 2krx (1/3)**
>
> We thank you for your thorough review and constructive feedback. In the rebuttal, we have tried to address each of your concerns point by point.
>
> > Weakness 1 (task may require multiple skills)
>
> We thank you for your observation. To quote from our current paper, “although individual questions may require multiple skills from each respective category”. We apologize if this was confusing and not sufficiently supported. To be precise, MMAU has 18% of questions that require more than one skill to solve and 8% of questions that require one skill each from information extraction and reasoning to solve. For better clarity, we have added more details to Appendix I and also added such examples in Table 12 of the Appendix. Below are a few examples for the purpose of the rebuttal:
>
> ```
> {
>         "question": "During what time frame can you hear the chord G# in the audio?",
>         "choices": [ "0.00 - 2.22", "2.22 - 4.44", "4.44 - 6.67", "6.67 - 8.89" ],
>         "answer": "0.00 - 2.22",
>         Skills involved: Harmony and Chord Progressions, Temporal Reasoning
> }
> {
>         "question": "What is the nature of the weather in the scene?",
>         "choices": ["Snowing heavily", "Sunny day", "Windy", "Heavy rain"],
>         "answer": "Heavy rain",
>         Skills involved:  Acoustic Scene Reasoning, Eco-Acoustic Knowledge
> }
> {
>        “question”: “What makes the last comment sarcastic in relation to the dialogue?”
>        “Choices”: [“Criticism of scientific method”, “Genuine admiration of intelligence”, “Requesting further explanation”, “Mocking exaggerated praise”]
>        “Skills Involved”: Multi Speaker Emotion Reasoning, Dissonant Emotion Interpretation
> }
> ```
> We have also included these examples in the Appendix I section of the revised paper.
>
> > Weakness 2 (source selection of data)
>
> Thank you for your question. In Section 3.2, we describe our data sourcing process, where we exclusively used audio from test sets or evaluation sets (when test sets were unavailable) to ensure unbiased and robust evaluation. Specifically:
> - **Music**: For music data, we utilized labeled test audio files, ensuring that the corresponding questions were highly relevant to the task at hand.
> - **Sound**: We utilized the evaluation set of AudioSet Strong, selecting audio clips containing a minimum of two and a maximum of five unique acoustic events, with each event lasting at least two seconds. This selection criterion ensured the inclusion of high-quality, distinguishable audio samples suitable for reasoning tasks.
> - **Speech**: For speech data, additional quality checks were applied by considering the lengths of transcriptions and ground truth labels, ensuring that the selected samples were clear and of adequate length for generating meaningful questions and answers.
>  These were preliminary checks conducted before the actual data filtering stage, which involved expert-driven refinement to ensure quality and diversity in the final selection of 10,000 data points. We have added this information in Appendix D.5 in the revised version of the paper.
>
> > Weakness 3 (quality check and bias control)
>
> Thank you for your question. Bias control in our annotation pipeline is implemented through a two-stage process:
> - **Filtering Stage**: As outlined in Section D. 1, we engage a diverse group of six expert annotators to mitigate individual biases and ensure consistency. Annotators follow a standardized set of guidelines, such as discarding short audio samples and those with mismatched labels, to establish a uniform understanding and baseline quality for the annotations.
> - **Expert Review Stage**: After the option augmentation process, each annotator independently scores questions generated by other experts on a 1-to-5 scale. Low scores are assigned to questions with misleading or overly correlated options, as well as those with incorrect answers. This scoring ensures the filtering of subpar samples and contributes to the reliability of the dataset.
>
> This structured, multi-stage approach ensures quality control and minimizes bias in the final selection of 10,000 samples in the MMAU dataset.

---

> ### Author Response · Authors · 2024-11-20
> **Response to Official Review of Submission3927 by Reviewer 2krx (2/3)**
>
> > Weakness 4 and 5 (open-ended vs MCQ evaluation)
>
> Thank You for the question! Our choice to use multiple-choice questions in MMAU is inspired by a wealth of prior research in vision [2] and language [1], which often rely on MCQ format for structured and scalable assessments.
> We would like to point out several reasons why we, and a plethora of prior work, have preferred MCQ-type questions:
> Open-ended responses require employing LLM-as-a-judge. While they allow evaluating more free-form responses by LALMs, this method faces several limitations including:
>
> (i) High evaluation costs for powerful judges, which are almost always proprietary
>
> (ii)  LLM-as-a-judge evaluation lacks standardization. Proprietary LLMs are taken down by their respective companies, and different researchers using different LLMs might lead to discrepancy in results due to different knowledge in them
>
> (iii) Lack of strong judges that can't (yet) take as input audio and return a faithful score for a response.
>
> By using MCQ-type questions, we not only identify the correct answer but also assess the model’s ability to eliminate plausible but incorrect distractors. This approach provides a more nuanced evaluation of the model’s reasoning capabilities, as it tests not just knowledge but also the process of reasoning through competing options.
> MCQ questions also allow the usage of standardized metrics like accuracy. This promotes fairness in the evaluation process.
>
> Finally, upon your request, we provide you a correlation between our regex matching and a more sophisticated GPT-based scoring of open-ended generations (more details in Appendix B.5) below:
>
> | Models | Our | GPT-4o |
> |-----|----|---|
> | SALMONN | 33.70 | 31.60  |
> | GAMA | 30.90 | 36.80  |
> | Qwen2-Audio-Instruct | 49.20 | 47.12 |
>
> As we can see, both scores are highly correlated, thus indicating the robustness of our regex evaluation protocol. We have also added this comparison in the revised version of the paper in Appendix B.5.
>
> **To also note, for reproducibility, we will open-source all codes. The current code is available in the Supplementary materials. Our evaluation script is adopted from well-established benchmarks MMLU [1] and MMMU [2].**
>
> > Question 1 (distinction between reasoning and extraction tasks)
>
> Thank you for this insightful question. The distinction between reasoning and extraction tasks was made to establish two broad categories for evaluating the skills of large audio-language models. This separation allows us to assess these models' capabilities individually on different skill sets, providing a structured framework for identifying their strengths and weaknesses. Such a framework is critical for guiding future research and enabling focused improvements in specific areas.
> As this is the first benchmark of its kind, we intentionally adopted this distinction to simplify the evaluation process and establish baseline performance metrics for LALMs across well-defined skill categories. We recognize that some tasks inherently require a combination of reasoning and extraction skills, and integrating such tasks is a natural next step for improving the comprehensive coverage of MMAU.
>
> Here are some examples of questions from our benchmark requiring both reasoning and extraction skills:
> ```
> {
>         "question": "During what time frame can you hear the chord G# in the audio?",
>         "choices": [ "0.00 - 2.22", "2.22 - 4.44", "4.44 - 6.67", "6.67 - 8.89" ],
>         "answer": "0.00 - 2.22",
>         Skills involved: Harmony and Chord Progressions, Temporal Reasoning
> }
> {
>         "question": "What is the nature of the weather in the scene?",
>         "choices": ["Snowing heavily", "Sunny day", "Windy", "Heavy rain"],
>         "answer": "Heavy rain",
>         Skills involved:  Acoustic Scene Reasoning, Eco-Acoustic Knowledge
> }
> ```
>
> > Question 2 (correct option generation in prompt)
>
> Thank You for the question! In Appendix L, the prompt is designed to instruct the model to generate contrasting options for each question, including the correct answer and distractor options. By providing the correct answer in the prompt, the model is tasked with formatting it alongside the distractor options, streamlining the process, and ensuring consistency in the generated output. This eliminates the need to add the correct answer later. To ensure reliability, we included a check to confirm that the correct answer was present in the generated options.

---

> > ### Author Response · Authors · 2024-11-20
> > **Response to Official Review of Submission3927 by Reviewer 2krx (3/3)**
> >
> > **References**
> >
> > [1] Hendrycks, Dan, et al. "Measuring massive multitask language understanding." arXiv preprint arXiv:2009.03300 (2020).
> >
> > [2] Yue, Xiang, et al. "Mmmu: A massive multi-discipline multimodal understanding and reasoning benchmark for expert agi." Proceedings of the IEEE/CVF Conference on Computer Vision and Pattern Recognition. 2024.

---

> > > ### Comment · Reviewer_2krx · 2024-11-22
> > > **Response to the authors**
> > >
> > > Thank you for your detailed response and the revised version of the paper. I appreciate the effort to improve the clarity and organization of your manuscript. Overall, the revisions have enhanced the readability and addressed some of my earlier concerns, but I still have a few questions and suggestions for further improvement:
> > >
> > > **Clarity and Presentation in the Main Text:**
> > > The main text still relies heavily on the appendix for clarification, which affects its standalone clarity. For instance, the justification for selecting 10,000 data points could be better elaborated directly in the main body. Similarly, newly added tables require improved formatting to ensure consistent presentation and readability. Additionally, there are areas where the writing could be further refined for better coherence and flow.
> > >
> > > **Question Regarding Experimental Setup:**
> > >
> > > For Question 1: I am still unclear about how you approached the Baseline tests. Specifically:
> > > Did you provide both the stems and answer choices of the multiple-choice questions to the model, or did you only provide the stems and ask it to generate the answers?
> > > How did you measure whether the Baseline's answers were correct?
> > > Could you elaborate on the implementation of the string match approach? A detailed explanation or pseudocode would be helpful.
> > >
> > > **Prompt Design Explanation:**
> > >
> > > For Question 2: Could you provide a concrete example to better illustrate your prompt design methodology? This would help clarify your design principles and assumptions.
> > >
> > > Overall, the paper has improved. I have raised my score in light of the progress made and look forward to your response and further refinements.

---

> > > > ### Author Response · Authors · 2024-11-22
> > > > **Response to Reviewer 2krx**
> > > >
> > > > > Clarity and Presentation in the Main Text
> > > >
> > > > Thank you for your insightful comments on the clarity and presentation of the main text.
> > > > To address your concerns, we have made the following improvements:
> > > > - We have added a detailed explanation in the main body of the paper (lines 214-230) about how we selected 10,000 data points from the 13 available audio corpora.
> > > > - We have included details about the question filtering process conducted by experts after option augmentation in the main text (lines 252-255).
> > > > - We have reformatted the newly added tables to ensure consistent presentation and better readability.
> > > >
> > > > We thank the reviewer for the valuable feedback, which has helped us strengthen the quality of our manuscript.
> > > >
> > > > > For Question 1: I am still unclear about how you approached the Baseline tests. Specifically: Did you provide both the stems and answer choices of the multiple-choice questions to the model, or did you only provide the stems and ask it to generate the answers? How did you measure whether the Baseline's answers were correct? Could you elaborate on the implementation of the string match approach? A detailed explanation or pseudocode would be helpful.
> > > >
> > > > **Ans.** We provided both the question stems and the multiple-choice answer choices to the LALM models. The models were instructed to select the correct choice from the given options.
> > > >
> > > > To determine whether the Baseline’s answers were correct, we implemented a string-matching evaluation. Here’s how the approach works:
> > > >
> > > > 1.	Tokenization: We break down both the model’s prediction and the correct answer into sets of lowercase words, ignoring case and punctuation differences.
> > > >
> > > > 2.	Evaluation Conditions:
> > > > - Condition 1: All tokens (words) from the correct answer must be present in the model’s prediction.
> > > > - Condition 2: The model’s prediction must not include tokens that are unique to the incorrect choices.
> > > >
> > > > If the model’s prediction satisfies both conditions, we consider it correct; otherwise, it’s marked incorrect. This method balances strict answer matching with flexibility for minor variations, ensuring a fair assessment of the model’s performance.
> > > > We have updated the relevant section in our paper (lines 351-353 and 356-357) and added a detailed explanation along with the pseudocode of the string-matching algorithm in Appendix I.
> > > >
> > > > > For Question 2: Could you provide a concrete example to better illustrate your prompt design methodology? This would help clarify your design principles and assumptions.
> > > >
> > > > **Ans.** Prompt: In Figure 10
> > > >
> > > > Input question details:
> > > > ```
> > > > {
> > > > 	“Question”: “What emotion is primarily conveyed by the string section in the audio?”,
> > > > 	“Question-Type”: "Emotional Tone Interpretation",
> > > > 	“Answer”: “Heroism”
> > > > }
> > > > ```
> > > > Prompt Output:
> > > > ```
> > > > {
> > > >   "question": "What emotion is primarily conveyed by the string section in the audio?",
> > > >   "question_type": "Emotional Tone Interpretation",
> > > >   "correct_answer": "Heroism",
> > > >   "options": "options": [ "Fear", "Despair", "Overconfidence", "Determination", "Elation", "Melancholy", "Heroism"]
> > > > }
> > > > ```
> > > > In this example, the model generates six contrastive options along with the correct answer. Including the correct answer doesn’t affect our overall method; it’s added to avoid extra processing steps. This demonstrates how we use strategies like opposites, partial correctness, and misleading similarities to create these options. By following these guidelines, we ensure the options are challenging and effectively test the user’s understanding.
> > > >
> > > > If we did not provide the correct answer to the model, the response looks like:
> > > > ```
> > > > {
> > > >   "QuestionType": "Emotional Tone Interpretation",
> > > >   "Question": "What emotion is primarily conveyed by the string section in the audio?",
> > > >   "CorrectAnswer": "Melancholy",
> > > >   "Options": ["Joy", "Contentment", "Despair", "Serenity", "Triumph", "Melancholy with a hint of hope"]
> > > > }
> > > > ```
> > > > However, these options may not accurately reflect the intended emotion, making them less effective for assessment. This is because the model might include irrelevant or incorrect emotions due to the assumption made without the correct answer.

---

> > > > > ### Author Response · Authors · 2024-11-25
> > > > > **Request to review the response**
> > > > >
> > > > > Dear reviewer 2krx,
> > > > >
> > > > > Thank you for taking the time to review our paper and thank you for the insightful engagement. Your comments have helped us improve the quality of the paper. This message is to request you to review our last response and the revised paper. We have made all changes suggested by you, responded to your questions and also made revisions to our paper to reflect the changes. Thank you once again for your valuable feedback!
> > > > >
> > > > > Best,
> > > > > Authors of Submission3927

---

> > > > > ### Comment · Reviewer_2krx · 2024-11-26
> > > > > **Response to the authors**
> > > > >
> > > > > I would like to express my gratitude to the authors for their revisions and the detailed responses provided. Based on your reply, I have carefully reviewed the paper and found that the areas of confusion have been adequately clarified. The explanation regarding character matching is also compelling.
> > > > >
> > > > > Regarding the prompt section, I would like to further inquire: if the answer is highly complex (rather than a single word), how does the LLM generate alternative answers? Additionally, in Figure 10, where the LLM is allowed to freely select 6 dimensions from a set of 10, can the generated answers be considered to follow a truly random distribution? My main concern stems from the fact that there does not appear to be an explicit check, nor any emphasis on ensuring the answers are randomly distributed within the prompt.
> > > > >
> > > > > Thank you once again for your responses. I believe some of my previous concerns have been addressed, and as a result, I have revised my score accordingly.

---

> > > > > > ### Comment · Reviewer_2krx · 2024-11-29
> > > > > > **Request to your response**
> > > > > >
> > > > > > Dear authors,
> > > > > >
> > > > > > Could you please answer my question? Since prompt quality control is very important. I hope to learn more about your design ideas or examples regarding complex scenes.

---

> > > > > > > ### Author Response · Authors · 2024-11-29
> > > > > > > **Response to Questions by Reviewer 2krx**
> > > > > > >
> > > > > > > Dear Reviewer 2krx,
> > > > > > >
> > > > > > > Thank You for your questions and thank you for your score increase! We truly appreciate the discussion which can improve the quality of our paper. We would like to respond to your additional question:
> > > > > > >
> > > > > > > > if the answer is highly complex (rather than a single word), how does the LLM generate alternative answers?
> > > > > > >
> > > > > > > **Ans.** Thank You for the question. Our prompt for option augmentation is constructed in a way that ensure that options more than a few words are handled appropriately. For example let us consider the following QA pair:
> > > > > > >
> > > > > > > ```
> > > > > > > {
> > > > > > > 'question': 'Based on the given audio, identify the source of the splintering sound.',
> > > > > > > 'choices': ['A tree falling', 'A wooden plank breaking', 'Glass shattering', 'Metal breaking'],
> > > > > > > 'answer': 'A wooden plank breaking',
> > > > > > > '}
> > > > > > > ```
> > > > > > > The contrastive options are ideally similar sounding words or concepts. This task of generating contrastive options is simple for an LLM, with its world knowledge and advanced linguistic comprehension capabilities, which is fed with the question and the ground-truth answer. Additionally, our prompt conditions (e.g., *Opposites* or * *Misleading Similarities* for this case) handles these cases by prompting the model to generate contrastive options according to the rules set in the prompt. Another compelling example is:
> > > > > > >
> > > > > > > ```
> > > > > > > {
> > > > > > >         "question": "What makes the last comment sarcastic in relation to the dialogue?",
> > > > > > >         "answer": "Mocking exaggerated praise.",
> > > > > > >         "choices": ["Criticism of scientific method.", "Genuine admiration of intelligence.", "Requesting further explanation.", "Mocking exaggerated praise."
> > > > > > >         ],
> > > > > > >     }
> > > > > > > ```
> > > > > > > Similarly for this example, our prompt rule *Topic Shift Confusion* and *Misleading Similarities* handles this case. **Our prompt was constructed with great care to handle long and short options, and the conditions are written to handle all cases.**
> > > > > > >
> > > > > > > > Additionally, in Figure 10, where the LLM is allowed to freely select 6 dimensions from a set of 10, can the generated answers be considered to follow a truly random distribution?
> > > > > > >
> > > > > > > **Ans.** Thank You for the question. Different options are meant for different modalities and for option augmentation, GPT might not use all dimensions for generating the 6 options. For example, while 3 is meant for speech questions, 4 is meant for general sound and music. Thus, we do not ensure a truly random distribution, which is analogous to prior research, MMMU [2], MMLU[1] and the MMLU-Pro [3].
> > > > > > >
> > > > > > > >  My main concern stems from the fact that there does not appear to be an explicit check, nor any emphasis on ensuring the answers are randomly distributed within the prompt.
> > > > > > >
> > > > > > > **Ans.** Thank You for the question. We generate 6 options and human annotators delete 2 low quality or redundant options for the final benchmark (lines 252 - 255). This is followed by randomizing `random.shuffle()` with different seeds by different annotators. These 2 steps ensure randomness in the answer position. This is inspired from MMLU and MMLU Pro, who do similar steps. *We do not perform any explicit check or do not re-andomize to ensure that the position of the true answer is well distributed among the 4 options.* We rather randomize the order of the options five times during inference and select the option chosen most frequently (mentioned in lines 358 - 359 of our paper). Both these steps are analogues to MMLU [3] in language and MMMU [2] in vision, where no explicit step was taken for answer order. In this step, we found only ~3.6% of the answers to have changed on randomizing the order, which shows that most models are robust to position bias. Since the task of the model is to just select the correct response, we might also consider i) an evaluation protocol by users that requires evaluation on 5 randomized orders, followed averaging and the requirement error bars. Additionally, we provide you with the distribution of answer positions for the output post GPT generated options:
> > > > > > >
> > > > > > > - Position 1: 21.07%
> > > > > > > - Position 2: 29.15%
> > > > > > > - Position 3: 23.52%
> > > > > > > - Position 4: 26.26%
> > > > > > >
> > > > > > > ### References.
> > > > > > > [1] https://arxiv.org/pdf/2406.01574.
> > > > > > > [2] https://arxiv.org/abs/2311.16502.
> > > > > > > [3] https://openreview.net/forum?id=d7KBjmI3GmQ.
> > > > > > >
> > > > > > > Please let us know if you have further queries and we would be happy to respond!
> > > > > > >
> > > > > > > Best,
> > > > > > > Authors of Submission3927

---

> > > > > > > > ### Author Response · Authors · 2024-12-02
> > > > > > > > **[Rebuttal period ending today] Request to review response and let us know if any further clarification is required**
> > > > > > > >
> > > > > > > > Dear reviewer 2krx,
> > > > > > > >
> > > > > > > > Thank you for taking the time to review our paper and thank you for the insightful engagement. Your comments have helped us improve the quality of the paper and we truly appreciate your engaging discussion and feedback. This message is to request you to review our last response and let us know if you have any further queries about our paper. Thank you once again for your valuable feedback!
> > > > > > > >
> > > > > > > > Best,
> > > > > > > > Authors of Submission3927

---

> ### Author Response · Authors · 2024-11-23
> **Request to review the response**
>
> Dear reviewer 2krx,
>
> Thank you for taking the time to review our paper and thank you for the insightful engagement. Your comments have helped us improve the quality of the paper. This message is to request you to review our last response and the revised paper. We have made all changes suggested by you, responded to your questions and also made revisions to our paper to reflect the changes. Thank you once again for your valuable feedback!
>
> Best,
> Authors of Submission3927

---

> > ### Author Response · Authors · 2024-11-24
> > **Request to review the response**
> >
> > Dear reviewer 2krx,
> >
> > Thank you for taking the time to review our paper and thank you for the insightful engagement. Your comments have helped us improve the quality of the paper. This message is to request you to review our last response and the revised paper. We have made all changes suggested by you, responded to your questions and also made revisions to our paper to reflect the changes. Thank you once again for your valuable feedback!
> >
> > Best,
> > Authors of Submission3927

---

### Official Review · Reviewer_ZoCS · 2024-10-20

**Soundness:** 3
**Presentation:** 3
**Contribution:** 3
**Rating:** 8
**Confidence:** 4

**Summary:**

The paper presents MMAU, a new benchmark to evaluate large audio-language models (LALMs) on complex multimodal audio understanding tasks. It includes 10,000 human-annotated audio-question-response pairs covering speech, sound, and music. The goal is to benchmark knowledge extraction and advanced reasoning across 27 distinct skills. The authors evaluate 18 open-source and proprietary models, revealing a significant gap between current LALMs and human capabilities. The benchmark's difficulty and breadth emphasize the limitations of current audio-language models, encouraging future research.

**Strengths:**

- MMAU significantly improves upon existing benchmarks by covering 27 distinct skills across three domains (speech, sound, and music). The low accuracy scores of state-of-the-art models highlight the benchmark's difficulty and push for more advanced models to handle complex tasks.
- It is the first benchmark for reasoning and expert-level knowledge extraction in literature, setting it apart from previous benchmarks focused on foundational audio processing tasks. This aligns with the growing demand for AI systems capable of human-like reasoning.
- The paper includes a detailed analysis of error types (perceptual, reasoning, etc.) across different ALMs and domains, offering valuable insights into current model limitations and how to improve them.
- The paper is well-written and easy to follow

**Weaknesses:**

- MMAU focuses solely on multiple-choice tasks, which could skew results towards models trained for MCQ-type question-answering and possibly even contrastive models. It would be beneficial to include an open-ended subset, even a small one, to contrast performance with the close-ended tasks.
- The current version treats skills needed for information extraction and reasoning as separate, potentially oversimplifying the evaluation of tasks requiring a combination of skills.
- [Minor] The contrastive models are tested on the MMAU dataset, but the results are added only in the appendix and Table 4. It would be better to add the Table 4 rows to the main Table 3 and include one main insight from the contrastive model results in the main paper.
- [Overall] The paper is well-executed, delivers on its promises, and devoid of any major weaknesses

**Questions:**

- The authors mention using robust regular expressions and response-processing workflows to extract key information, matching it to the provided options via string matching. Evaluating ALMs that produce open-ended responses is challenging as they often don't adhere to multiple-choice options or instructions, making regex unsuitable. Have the authors considered using an LLM to evaluate outputs for this task? More details on the regex method, including human annotation verification for a subset of randomly sampled data, would be helpful.
- In Appendix E, the authors list dataset sources used. Among these, AudioSet and AudioSet-strong are commonly used to train ALMs. This overlap between training and test audio data could skew results and conclusions. Have the authors conducted experiments to ensure this isn't the case or used a separate test set to avoid this issue? Clarification would be appreciated.
- Additional information on the synthetic data generation and its domain would be helpful, I could not find this in the paper

---

> ### Author Response · Authors · 2024-11-19
> **Response to Official Review of Submission3927 by Reviewer ZoCS (1/2)**
>
> We thank you for your thorough review and constructive feedback. We have tried to address each of your concerns point by point in the rebuttal.
>
> ### Weaknesses
>
> > Weakness 1 (About MCQ Questions)
>
> Ans. Thank you for your suggestion regarding open-ended questions. Our decision to use multiple-choice questions (MCQs) in MMAU is motivated by extensive prior research in vision [6,8] and language [5,7], where MCQ formats are widely used for structured and scalable evaluations.
>
> Here are several reasons why we, along with much of prior work, prefer MCQ-type questions:
>
> - Open-ended responses typically require using an LLM-as-a-judge, which presents several limitations: (i) The most powerful (and widely used) judges, such as GPT-4 or Gemini Pro, are proprietary and expensive to use. (ii) Proprietary LLMs may be retired by their companies, and using different LLMs as judges can lead to inconsistent results due to differences in knowledge and behavior. (iii) There are no sufficiently robust LLMs capable of evaluating audio-based inputs and returning reliable scores for open-ended responses.
>
> - MCQs allow us to not only identify the correct answer but also assess the model’s ability to distinguish between plausible but incorrect distractors. This provides a more nuanced evaluation of the model’s reasoning capabilities, testing both knowledge and the process of reasoning through competing options.
>
> - MCQs enable the use of standardized metrics like accuracy, which ensures fairness and comparability in the evaluation process across models and research efforts.
>
> **However, in light of the request, we have planned to add a subset of 500 QA pairs in the final version of MMAU post acceptance. We apologize for not being able to accomplish this in the  short period of rebuttal but promise to do this in the final version.**
>
> > Weakness 3 (About scores for Audio -Language Encoders)
>
> **Ans.** Thank You for the question! The primary reason for separating the results of contrastive Audio-Language Encoders (ALEs), such as CLAP, is that their evaluation strategy differs significantly from that of LALMs, as detailed in Table 3. As explained in Appendix Section B.1, ALEs inherently lack instruction-following capabilities. Consequently, their evaluation requires an ad hoc method similar to the approach described by Deshmukh et al. (2024). In this method, each question-choice pair is transformed into a hypothesis using GPT-4. The audio and hypotheses are then encoded using ALEs, and the hypothesis with the highest cosine similarity to the audio embedding is selected as the answer.
>
> We appreciate the reviewer’s suggestion to summarize the key findings of the ALE experiments in the main paper. We have added this detail to Section 5.1 of our paper, where we provide a concise summary of ALE performance on MMAU.
>
> ### Questions
>
> > Question 1 (About regular expressions for evaluation)
>
> **Ans.** Thank You for the question. Our regex method to evaluate responses has been inspired by well-established benchmarks like MMLU[1] and MMMU[3].
> In addition, our evaluation protocol is standard across benchmarks[1,3,4,5]
>
> At your request, we provide you a correlation between our regex matching and GPT scores below:
> | Models | Our | GPT-4o |
> |-----|----|---|
> | SALMONN | 33.70 | 31.60  |
> | GAMA | 30.90 | 36.80  |
> | Qwen2-Audio-Instruct | 49.20 | 47.12  |
>
> As we can see, both scores are highly correlated, thus indicating the robustness of our regex evaluation protocol. We have also added this comparison in the revised version of the paper in Appendix B.5.
>
> > Question 2 (About AudioSet strong and possible data leakage)
>
> **Ans.** Thank You for your question. All questions for MMAU for all datasets were sourced from the test set of all datasets. For AudioSet strong, we employ the eval set. Thus, we can confirm there is no data leakage for any dataset used to evaluate MMAU. We have added this detail to Section 3.2 of our paper.
>
> > Question 3 (About synthetic data generated)
>
> **Ans.** We synthesized audio data only for tasks where no open-source audio datasets were available. For audio generation, we utilized Text-to-Audio models such as Parler-TTS for speech and Stable Audio for other audio types.
>
> For instance, in the task of Multi-Speaker Role Mapping, no open-source speech datasets explicitly provide information about the roles of speakers in a conversation. To address this gap, we collaborated with domain experts to design synthetic textual conversations that clearly define each speaker's role (e.g., doctor, patient, etc.). These conversations were then converted into speech using text-to-audio models. Finally, domain experts reviewed the generated speech and curated reasoning-based question-answer pairs tailored to the specific skill being assessed.
>
> **continued**

---

> ### Author Response · Authors · 2024-11-19
> **Response to Official Review of Submission3927 by Reviewer ZoCS (2/2)**
>
> The list of tasks for which synthetic data was generated, along with their respective counts, is as follows:
>
> | **Category** | **Tasks**                          | **Count** |
> |--------------|------------------------------------|-----------|
> | **Sound**    | Temporal Event Reasoning           | 250       |
> | **Sound**    | Ambient Sound Understanding        | 436       |
> | **Speech**   | Event-Based Knowledge Retrieval    | 316       |
> | **Speech**   | Multi Speaker Role Mapping         | 260       |
> | **Speech**   | Phonological Sequence Understanding| 150       |
>
> Detailed information about the synthetic data generation process has been added in Appendix C.
>
> > Weakness 2 (About distinction between Information extraction and reasoning)
>
> **Ans.** We appreciate the reviewer’s concern and acknowledge the limitation of treating the skills required for information extraction and reasoning as entirely separate sets. As highlighted in the limitations section, the current version of MMAU does not include *many* questions that require a combination of these skills. We recognize that incorporating such overlapping skills is crucial for a more comprehensive evaluation and plan to address this in future benchmark iterations.
>
> However, MMAU has few questions (~8%) with skills needed from both information extraction and reasoning. We present a few below:
>
> ```
> {
>         "question": "During what time frame can you hear the chord G# in the audio?",
>         "choices": [
>             "0.00 - 2.22",
>             "2.22 - 4.44",
>             "4.44 - 6.67",
>             "6.67 - 8.89"
>         ],
>         "answer": "0.00 - 2.22",
>         Skills involved: Harmony and Chord Progressions, Temporal Reasoning
> }
>
> {
>         "question": "What is the nature of the weather in the scene?",
>         "choices": [
>             "Snowing heavily",
>             "Sunny day",
>             "Windy",
>             "Heavy rain"
>         ],
>         "answer": "Heavy rain",
>         Skills involved:  Acoustic Scene Reasoning, Eco-Acoustic Knowledge
> }
> ```
> We have added this detail to the Appendix I section of the revised version of our paper.
> ### References:
>
> [1] Hendrycks, Dan, et al. "Measuring massive multitask language understanding." arXiv preprint arXiv:2009.03300 (2020).
> [2] Meng, Fanqing, et al. "Mmiu: Multimodal multi-image understanding for evaluating large vision-language models." arXiv preprint arXiv:2408.02718 (2024).
> [3] Yue, Xiang, et al. "Mmmu: A massive multi-discipline multimodal understanding and reasoning benchmark for expert agi." Proceedings of the IEEE/CVF Conference on Computer Vision and Pattern Recognition. 2024.
> [4] Weck, Benno, et al. "Muchomusic: Evaluating music understanding in multimodal audio-language models." arXiv preprint arXiv:2408.01337 (2024).
> [5] Yang, Qian, et al. "AIR-Bench: Benchmarking Large Audio-Language Models via Generative Comprehension." arXiv preprint arXiv:2402.07729 (2024).

---

> > ### Comment · Reviewer_ZoCS · 2024-11-23
> > **Reviewer response**
> >
> > Thanks for the detailed response for all the questions and concerns reported
> >
> > The author's response addressed my questions and any follow up questions I had in global comment. I will maintain my score and recommend accepting this paper to AC.

---

### Official Review · Reviewer_hLoT · 2024-10-28

**Soundness:** 3
**Presentation:** 4
**Contribution:** 3
**Rating:** 8
**Confidence:** 5

**Summary:**

This work introduces MMAU, an audio understanding benchmark designed to evaluate the advanced auditory perceptual and reasoning capabilities of LALMs. MMAU features 10,000 human-annotated audio-question-response pairs, encompassing 27 distinct skills across unique tasks. Most existing LALMs are tested on MMAU in the paper, highlighting the challenges in auditory understanding faced by current models.

**Strengths:**

- In this paper, a large-scale audio understanding benchamrk "MMAU" is built to evaluate LALMs. Different from previous benchmarks, MMAU not only pays more attention to deeper and more difficult auditory reasoning tasks, but also covers a wide range of sound signals as well.
- I believe MMAU is a reliable benchmark, as it is carefully designed during the build process with human review at each step.
- Most exsiting LALMs are evaluated on MMAU in this paper. Besides, the authors have analysed the weaknesses of existing LALMs in detail, providing directions for further studies.

**Weaknesses:**

- It seems that MMAU primarily focuses on short audio (around 10 seconds) and lacks evaluations involving perception and reasoning for long audio. Since long audio includes more contextual information, the model's ability to understand long audio might offer a broader indication of its overall performance.
- MMAU is still predominantly multiple choice, but existing LALMs may not be good at multiple choice. Perhaps open-ended questions could be used as prompts instead when testing.

**Questions:**

- What tasks is the synthesised data primarily aimed at? How is the data synthesised?
- How is the difficulty ("easy", "medium", "hard") of a question determined? Since for some skills, existing models consistently perform badly regardless of the difficult levels of the questions, does this mean that the difficulty should be classified by the skill type, rather than the question?

---

> ### Author Response · Authors · 2024-11-19
> **Response to Official Review of Submission3927 by Reviewer hLoT (1/2)**
>
> We thank you for your thorough review and constructive feedback. We have tried to address each of your concerns point by point in the rebuttal.
>
> ### Weaknesses
>
> > Weakness 1 (About long audios)
>
> **Ans.** Thank you for raising this important question! We fully acknowledge that long audio could provide a broader evaluation of LALM performance. However, most current open-source and open-access Large Audio-Language Models (LALMs), such as LTU (10 seconds), GAMA (10 seconds), SALMONN (10 seconds), and Qwen2-Audio (30 seconds), are limited to processing short audio inputs. This limitation also stems from the majority of training data available for these models, as well as the audio encoders, which typically support inputs of 30 seconds or less.
>
> Given these constraints, we have focused the current version of MMAU on reasoning over short audio clips, leaving long audio processing outside the current scope. However, we recognize the importance of this area and plan to include long audio reasoning as an immediate goal for future versions of MMAU. This approach is analgous in video understanding research, where initial efforts centered on short video benchmarks [1,2], eventually transitioning to long video understanding [3,4] as more capable long video models emerged [5,6]. We believe similar advancements will occur in the audio domain as the research community continues to push LALM capabilities.
>
> Following are the average durations for MMAU:
>
> | Task      | Mean Duration | Min. | Max. |
> |------------|---------|---------|----------|
> | Speech | 14.82 | 5.07   | 49.58  |
> | Sound   | 11.80 | 3.08   | 30.00  |
> | Music    | 16.03 | 4.35   | 30.11  |
>
> We have included additional details and discussion in the Limitations and Future Works section and updated Table 1 of our paper.
>
> > Weakness 2 (About MCQ)
>
> **Ans.** Thank you for your suggestion regarding open-ended questions. Our decision to use multiple-choice questions (MCQs) in MMAU is motivated by extensive prior research in vision [6,8] and language [5,7], where MCQ formats are widely used for structured and scalable evaluations.
>
> Here are several reasons why we, along with much of prior work, prefer MCQ-type questions:
>
> - Open-ended responses typically require using an LLM-as-a-judge, which presents several limitations: (i) The most powerful (and widely used) judges, such as GPT-4 or Gemini Pro, are proprietary and expensive to use. (ii) Proprietary LLMs may be retired by their companies, and using different LLMs as judges can lead to inconsistent results due to differences in knowledge and behavior. (iii) There are no sufficiently robust LLMs capable of evaluating audio-based inputs and returning reliable scores for open-ended responses.
>
> - MCQs allow us to not only identify the correct answer but also assess the model’s ability to distinguish between plausible but incorrect distractors. This provides a more nuanced evaluation of the model’s reasoning capabilities, testing both knowledge and the process of reasoning through competing options.
>
> MCQs enable the use of standardized metrics, such as accuracy, which ensures fairness and comparability in the evaluation process across models and research efforts.
>
> ### Questions
>
> > Question 1 (On synthetic data)
>
> **Ans.** We synthesized audio data only for tasks where no open-source audio datasets were available. For audio generation, we utilized Text-to-Audio models such as Parler-TTS for speech and Stable Audio for other audio types.
>
> For instance, in the task of Multi-Speaker Role Mapping, no open-source speech datasets explicitly provide information about the roles of speakers in a conversation. To address this gap, we collaborated with domain experts to design synthetic textual conversations that clearly define each speaker's role (e.g., doctor, patient, etc.). These conversations were then converted into speech using text-to-audio models. Finally, domain experts reviewed the generated speech and curated reasoning-based question-answer pairs tailored to the specific skill being assessed.
>
> The list of tasks for which synthetic data was generated, along with their respective counts, is as follows:
>
> | **Category** | **Tasks**                          | **Count** |
> |--------------|------------------------------------|-----------|
> | **Sound**    | Temporal Event Reasoning           | 250       |
> | **Sound**    | Ambient Sound Understanding        | 436       |
> | **Speech**   | Event-Based Knowledge Retrieval    | 316       |
> | **Speech**   | Multi Speaker Role Mapping         | 260       |
> | **Speech**   | Phonological Sequence Understanding| 150       |
>
> Appendix C of the revised version of our paper now includes detailed information about the synthetic data generation process.
>
> **Continued**

---

> > ### Author Response · Authors · 2024-11-19
> > **Response to Official Review of Submission3927 by Reviewer hLoT (2/2)**
> >
> > ### References
> >
> > [1] Li, B., Wang, R., Wang, G., Ge, Y., Ge, Y., & Shan, Y. (2023). Seed-bench: Benchmarking multimodal llms with generative comprehension. arXiv preprint arXiv:2307.16125.
> > [2] Pătrăucean, V., Smaira, L., Gupta, A., Continente, A. R., Markeeva, L., Banarse, D., ... & Carreira, J. (2023). Perception test: A diagnostic benchmark for multimodal video models. arXiv preprint arXiv:2305.13786.
> > [3] Zhou, J., Shu, Y., Zhao, B., Wu, B., Xiao, S., Yang, X., ... & Liu, Z. (2024). MLVU: A Comprehensive Benchmark for Multi-Task Long Video Understanding. arXiv preprint arXiv:2406.04264.
> > [4] Chandrasegaran, K., Gupta, A., Hadzic, L. M., Kota, T., He, J., Eyzaguirre, C., ... & Fei-Fei, L. (2024). HourVideo: 1-Hour Video-Language Understanding. arXiv preprint arXiv:2411.04998..
> > [5] Xue, F., Chen, Y., Li, D., Hu, Q., Zhu, L., Li, X., ... & Han, S. (2024). Longvila: Scaling long-context visual language models for long videos. arXiv preprint arXiv:2408.10188.
> > [6] Shen, X., Xiong, Y., Zhao, C., Wu, L., Chen, J., Zhu, C., ... & Chandra, V. (2024). LongVU: Spatiotemporal Adaptive Compression for Long Video-Language Understanding. arXiv preprint arXiv:2410.17434.
> > [5] Hendrycks, Dan, et al. "Measuring massive multitask language understanding." arXiv preprint arXiv:2009.03300 (2020).
> > [6] Meng, Fanqing, et al. "Mmiu: Multimodal multi-image understanding for evaluating large vision-language models." arXiv preprint arXiv:2408.02718 (2024).
> > [7] Hendrycks, Dan, et al. "Measuring massive multitask language understanding." arXiv preprint arXiv:2009.03300 (2020).
> > [8] Yue, Xiang, et al. "Mmmu: A massive multi-discipline multimodal understanding and reasoning benchmark for expert agi." Proceedings of the IEEE/CVF Conference on Computer Vision and Pattern Recognition. 2024.

---

> > > ### Comment · Reviewer_hLoT · 2024-11-22
> > > **Response to the authors**
> > >
> > > Thanks for the response.
> > >
> > > Do the authors have an answer for Question 2?

---

> ### Author Response · Authors · 2024-11-22
> **Response to Reviewer hLoT**
>
> We apologize for missing your question earlier. Amidst many queries, we updated the relevant section in the paper (Appendix D.4) but missed addressing it in our response. Here is our response to Question 2:
>
> > Question 2 (determining difficulty of a question)
>
> **Ans.** As mentioned in Section 3.1, each question in MMAU requiring specific skills is manually categorized by domain experts into easy, medium, or hard difficulty levels. Our domain experts rated the difficulty of each question from 1-10. For each question, we then averaged the scores from each expert and assigned the question to be “easy” if the score was between 1-3, medium if 4-6, and hard if > 6. The score for the difficulty level was assigned based on the level of expertise or information required to respond correctly to a question. We have added this detail to Appendix D.4 of our paper.

---

> > ### Comment · Reviewer_hLoT · 2024-11-22
> > **Response to the authors**
> >
> > Thanks for the detailed response.
> >
> > The author's response addressed my questions well. I will maintain my previous score and recommend accepting this paper.

---

> ### Comment · Reviewer_hLoT · 2024-11-22
> **Response to the authors**
>
> Thanks for the authors' reply. Could you please give some examples of questions with different difficulties? Also, how do humans perform on questions of varying difficulties? Besides, for humans, is there a similar phenomenon to that of models, whereby they perform poorly on certain types of tasks?

---

> > ### Author Response · Authors · 2024-11-22
> > **Response to Reviewer hLoT**
> >
> > > Could you please give some examples of questions with different difficulties?
> >
> > **Ans.** Here are two examples of questions from each difficulty level:
> > ```
> > {
> >         "question": "Which emotion is most likely to be evoked by the audio?",
> >         "choices": [ "Sadness and grief",  "Calm and tranquility",  "Youthful and energetic", "Fear and anxiety"],
> >         "answer": "Youthful and energetic",
> >         skills involved: "Emotional Tone Interpretation",
> >         "difficulty": "easy"
> > },
> > {
> >         "question": "What emotion is primarily conveyed by the string section in the audio?",
> >         "choices": ["Sadness",  "Joy", "Tension", "Heroism"],
> >         "answer": "Heroism",
> >         skills involved : "Emotional Tone Interpretation",
> >         "difficulty": "medium"
> > },
> > {
> >         "question": "What is a likely reason for the audio not being very pleasant to listen to?",
> >         "choices": [ "Monotonous rhythm",  "Heavy elements",  "Lack of melody",  "Slow tempo" ],
> >         "answer": "Heavy elements",
> >         skills involved : "Emotional Tone Interpretation",
> >         "difficulty": "hard"
> > },
> > ```
> > ```
> > {
> >         "question": "How many times does the word 'train' appear in the sentence?",
> >          "choices": [ "three",  "two", "one", "five" ],
> >          "answer": "one",
> >         "Skills involved": "Phonological Sequence Decoding",
> >         "difficulty": "easy",
> > },
> > {
> >         "question": "What did First speaker tell Second speaker not to look at?",
> >         "choices": [ "Her left hand.", "Her right hand.", "Her face.", "Her left foot." ],
> >         "answer": "Her left hand.",
> >        "difficulty": "medium",
> >         "Skills involved": "Conversational Fact Retrieval"
> > },
> > {
> >         "question": "Did any speaker's emotion shift during the conversation?",
> >         "choices": [ "second speaker",  "none of the speakers", "first speaker",  "both speakers"],
> >         "answer": "both speakers",
> >         "difficulty": "hard",
> >         "Skills involved": "Emotion Flip Detection"
> > },
> > ```
> >
> > > Also, how do humans perform on questions of varying difficulties?
> >
> > **Ans.** Human eval was conducted on the test-mini subset of the benchmark. Here is the human-eval performance on different difficulties:
> >
> > |  Test-mini | Easy | Medium | Hard |
> > |-----------|------|--------|------|
> > | Human-eval | 92.4 | 84.9   | 71.3 |
> >
> > The table contains the averaged performance of 8 human participants on different difficulty categories of the test-mini subset of the benchmark.
> >
> > > Besides, for humans, is there a similar phenomenon to that of models, whereby they perform poorly on certain types of tasks?
> >
> > **Ans.** Yes, humans also face challenges with certain tasks, similar to models. For example, socio-cultural interpretation can be difficult, with human performance ranging between 68% and 85%. However, humans outperform models in temporal reasoning by ~52%. On the other hand, tasks like exact chord progressions and phonological sequence decoding are especially tough for humans, with scores of 26% and 42%, respectively.

---

### Official Review · Reviewer_W76o · 2024-11-03

**Soundness:** 3
**Presentation:** 3
**Contribution:** 4
**Rating:** 8
**Confidence:** 5

**Summary:**

This paper introduces MMAU, a new benchmark designed for audio large language models, akin to the role of MMLU and MMMU for text and vision. The authors hired "domain experts" to curate a challenging set of 10,000 annotated audio-based question-response pairs spanning speech, music, and environmental sounds. The author further benchmarked 18 open-source and proprietary large audio-language models, and shows a few interesting observation and insights.

**Strengths:**

1. The topic of this paper is very important — currently, most audio LLMs are benchmarked using different evals, and researchers must test other models for comparison, which is tedious and impractical for many. A unified benchmark is common in text and vision, such as MMLU, MMMU, and DocVQA, and has already demonstrated its effectiveness as a good measure of models. We need one for audio.

2. The authors also benchmarked 18 LLMs, which is a non-trivial task as some open-source repositories are not easy to use. With this benchmark, we gain insights into performance comparisons among them, making this paper also serve as a very valuable review paper. The result numbers contain a lot of information.

3. The paper includes some good discussions.

4. Overall, the writing is clear, with detailed information provided in the appendix.

**Weaknesses:**

1. I would like to see more discussion for an ICLR submission. Although the authors did significant work, I encourage them to add more insights to the results, such as exploring why one model performs better than another in specific benchmarks.
- e.g., is the difference of a specific test more likely from the training data (e.g., Google's model trained on more data than those from academia), or from the architecture/training (e.g., early fusion, late fusion, discrete token vs continuous embedding vs text input). And what's the potential way to improve.

2. Evaluations are crucial but must be carefully designed. I did not notice a few-shot setting mentioned in the paper (unless I overlooked it), yet few-shot evaluation is quite common in LLM benchmarking (i.e., Q1->A1; Q2->A2; Q3-?).
- If running few shot on all models / tests is hard, I suggest to run it for some representative models and report the gap between zero-shot and few-shot, this is particular important for model not post-trained on things like multi-choice questions (select one from ABCD).

3. I understand this might be due to the timing of when the authors prepared the paper or potential rate limitations, so this is not a major flaw. However, given that OpenAI now offers real-time API and voice mode API, it would be a valuable addition to include these capabilities.
- And I understand it might be quite expensive to run the full test, if that is the case, maybe you can run on a set of hard tests and compare with Google's model. But I feel a full run would greatly increase the value of the paper - it tells the reader which is the best audio model.

**Questions:**

Section 5.2

I am wondering about the breakdown (e.g., categories) for this experiment. Do you provide the ground truth text of the speech to the model? It seems a bit strange to me that, for sound classification and speech recognition, the model could do anything meaningful with Gaussian noise.

---

> ### Author Response · Authors · 2024-11-19
> **Response to Official Review of Submission3927 by Reviewer W76o (1/2)**
>
> We thank you for your thorough review and constructive feedback. We have tried to address each of your concerns point by point in the rebuttal.
>
> ### Weaknesses
>
> > Weakness 1
>
> **Ans.** We appreciate your recognition of the significance of our work. Section 5.1 provides a discussion on the performance comparison of different LALMs, and we would like to take this opportunity to elaborate further:
>
> - Model size is a crucial factor influencing performance on MMAU. Larger LLMs inherently store more knowledge and demonstrate superior reasoning capabilities. As a result, LALMs incorporating larger LLMs perform better. For instance, LTU, which has 6 billion fewer parameters than SALMONN, shows an average performance drop of 14% in Table 3.
> - Models trained for specific domains outperform general-purpose models on corresponding tasks. For example, M2UGen, designed for music understanding and reasoning, surpasses general-purpose models like LTU and GAMA by up to 15% on music-related tasks.
> - Proprietary and open-access models outperform fully open-source models across all tasks. This suggests that larger datasets and additional training resources likely contribute to enhanced audio perception and reasoning performance on MMAU.
>
> We believe these insights will further strengthen the significance of our findings and we have added these details to the revised version of our paper.
>
> > Weakness 2
>
> **Ans.** Thank you for your question! We did not conduct few-shot (or in-context learning) experiments because most openly available Large Audio-Language Models lack support for multiple audio inputs, a feature essential for enabling few-shot learning. As a result, comparing the limited number of models that support this functionality would have been challenging. However, based on your request, we are sharing results for Qwen2-Audio, which does support multiple audio inputs. Below are the results on the test-mini subset:
>
> | Qwen 2 Audio Instruct | Domain Accuracy         |     |  | Difficulty Accuracy      |  |   | Total |
> |----|----|-----|-----|-----|----|----|-------|
> |                        | Sound                  | Music                  | Speech                 | Easy                    | Medium                 | Hard                   |       |
> | **0 shot**             | 54.95                  | 50.98                  | 42.04                  | 36.05                   | 59.80                  | 41.81                  | 49.2  |
> | **1 Shot**             | 51.95                  | 45.21                  | 31.83                  | 27.13                   | 53.92                  | 36.64                  | 43.0  |
> | **3 Shot**             | 14.41                  | 26.35                  | 14.11                  | 14.73                   | 23.53                  | 10.78                  | 18.3  |
> | **5 Shot**             | 16.52                  | 25.45                  | 23.12                  | 14.34                   | 25.10                  | 22.41                  | 21.7  |
>
> *Performance comparison of Qwen2-Audio on test-mini across different few-shot settings.*
>
> We have also added these results to the revised version of our paper in Appendix B.3.
>
> > Weakness 3
>
> **Ans.** Thank you for your question! You are correct that OpenAI's audio APIs were not available at the time of paper submission. However, upon your request, we explored the recently launched OpenAI audio APIs and found that OpenAI currently provides two ways to interact with GPT models using voice inputs:
>
> - **Real-Time Mode**: This mode supports speech-to-speech interactions and can also handle text inputs and text outputs. However, it does not allow simultaneous use of text and audio inputs.
>
> - **Voice Mode**: This mode extends GPT inputs to include audio. Unfortunately, this feature is only accessible to GPT Plus users through the ChatGPT UI and does not have a public API available. Given the size of our benchmark, evaluation through the UI is not feasible.
>
> Due to these limitations, we regret that we are unable to provide GPT scores at this time. However, we will monitor future developments and update our paper for the camera-ready version (if accepted) or in future revisions if OpenAI releases an API that enables evaluation on MMAU.
>
> ### Questions
>
> > Question 1
>
> **Ans.** The experiment depicted in Figure 5 aims to investigate whether LALMs genuinely attend to audio inputs when answering complex MCQ questions in the MMAU benchmark. In this setup, we present the average accuracy scores across all tasks. Under the noisy condition, the models are not provided with the ground truth audio captions. To ensure the model responds, we explicitly instruct it to select an answer for each question.
>
> **continued**

---

> ### Author Response · Authors · 2024-11-19
> **Response to Official Review of Submission3927 by Reviewer W76o (2/2)**
>
> Below we provide the task-wise breakdown of the scores for the experiment in Section 5.2:
>
> | Skills                             | GAMA (Audio) | GAMA (Noise) | SALMONN (Audio) | SALMONN (Noise) |
> |------------------------------------|--------------|--------------|-----------------|-----------------|
> | Acoustic Scene Reasoning           | 0.34         | 0.28         | 0.43            | 0.44            |
> | Conversational Fact Retrieval      | 0.31         | 0.14         | 0.31            | 0.30            |
> | Harmony and Chord Progressions     | 0.20         | 0.18         | 0.23            | 0.23            |
> | Instrumentation                    | 0.23         | 0.14         | 0.30            | 0.32            |
> | Phonological Sequence Understanding| 0.24         | 0.06         | 0.03            | 0.03            |
>
> Overall, we observe that:
> - Our benchmark is robust as most models perform close to random chance when addition of white noise.
> - When LALMs *do not undergo performance degradation with white noise*, their scores are ideally close to random chance with and without white noise.
> - LALMs like SALMONN and MuLLaMa exhibit minimal performance degradation with white noise. This suggests that these models may be **guessing randomly** or **relying on strong language priors** from their LLM counterparts to select an answer.
>
> We have added the complete Table and discussion to the Appendix B.4 of our paper.

---

> ### Author Response · Authors · 2024-11-23
> **Request to review the rebuttal**
>
> Dear reviewer W76o,
>
> Thank you for taking the time to review our paper. We have addressed your concerns in our submitted response and provided a revised version of the paper. As the rebuttal period is nearing its conclusion, we kindly request you to review our rebuttal and share any additional comments or concerns you may have. Thank you once again for your valuable feedback!
>
> Best,
> Authors of Submission3927

---

> ### Author Response · Authors · 2024-11-24
> **Request to review the rebuttal**
>
> Dear reviewer W76o,
>
> Thank you for taking the time to review our paper. We have addressed your concerns in our submitted response and provided a revised version of the paper. As the rebuttal period is nearing its conclusion, we kindly request you to review our rebuttal and share any additional comments or concerns you may have. Thank you once again for your valuable feedback!
>
> Best,
> Authors of Submission3927

---

> > ### Author Response · Authors · 2024-11-25
> > **Request to review the rebuttal**
> >
> > Dear reviewer W76o,
> >
> > Thank you for taking the time to review our paper. We have addressed your concerns in our submitted response and provided a revised version of the paper. As the rebuttal period is nearing its conclusion, we kindly request you to review our rebuttal and share any additional comments or concerns you may have. Thank you once again for your valuable feedback!
> >
> > Best,
> > Authors of Submission3927

---

> > > ### Author Response · Authors · 2024-11-28
> > > **Request to review the rebuttal**
> > >
> > > Dear reviewer W76o,
> > >
> > > Thank you for taking the time to review our paper. We have addressed your concerns in our submitted response and provided a revised version of the paper. As the rebuttal period is nearing its conclusion, we kindly request you to review our rebuttal and share any additional comments or concerns you may have. Thank you once again for your valuable feedback!
> > >
> > > Best,
> > > Authors of Submission3927

---

> > > > ### Comment · Reviewer_W76o · 2024-12-02
> > > > **thanks for the response and improvement**
> > > >
> > > > The explaination and new results improves the pape. I changed my score from 6 to 8. I appreicate the authors' effort.

---

> > > > > ### Author Response · Authors · 2024-12-02
> > > > > **Thank You!**
> > > > >
> > > > > Thank You for your response and your time spent in reviewing our paper and rebuttal. Your feedback is invaluable to us and will help us improve the quality of the paper!
> > > > >
> > > > > Best,
> > > > > Authors of Submission3927

---

### Author Response · Authors · 2024-11-21
**General Response to All Reviewers and Request to Review the Rebuttal**

Dear Reviewers,

We thank the reviewers for their insightful and positive feedback! We are encouraged to find that all reviewers find our MMAU benchmark a valuable contribution to the audio community. We are also happy to know that most reviewers found our paper containing sufficient details and good experiments (W76o, hLoT, ZoCS) and overall well presented (all reviewers).

In our rebuttal, we have addressed each concern by each reviewer in detail, point-by-point, and have updated a revised version of the paper with all these details. Most reviewers required extra details about the benchmark and experiments and did not find any significant flaws with the benchmark or experimental setup.

One common concern is our evaluation metric and the the choice of MCQ questions for MMAU to evaluate LALMs which are better at free-form generation. To this question, we would like to point out that our choice of MCQ and regex based evaluation (with the preferred metric as accuracy) **has been inspired from a wealth of prior art and established benchmarks in the AI community [1,2] (and many such benchmarks).** Adding open-ended QAs with different evaluation scheme is a part of future work we would like address in future work. Finally, two additional points to support our choice:

- Open-ended responses typically require using an LLM-as-a-judge, which presents several limitations: (i) The most powerful (and widely used) judges, such as GPT-4 or Gemini Pro, are proprietary and expensive to use. (ii) Proprietary LLMs may be retired by their companies, and using different LLMs as judges can lead to inconsistent results due to differences in knowledge and behavior. (iii) There are no sufficiently robust LLMs capable of evaluating audio-based inputs and returning reliable scores for open-ended responses.

- MCQs allow us to not only identify the correct answer but also assess the model’s ability to distinguish between plausible but incorrect distractors. This provides a more nuanced evaluation of the model’s reasoning capabilities, testing both knowledge and the process of reasoning through competing options.

**Finally, we would request all reviewers to please go through the rebuttal, our responses and let us know if they have more questions. We would be more than happy to respond to any more questions.**

Best,
Authors of #Submission3927





### References
[1] Hendrycks, Dan, et al. "Measuring massive multitask language understanding." arXiv preprint arXiv:2009.03300 (2020).

[2] Yue, Xiang, et al. "Mmmu: A massive multi-discipline multimodal understanding and reasoning benchmark for expert agi." Proceedings of the IEEE/CVF Conference on Computer Vision and Pattern Recognition. 2024.

---

### Meta-Review · Area_Chair_kg71 · 2024-12-20

**Metareview:**

This paper introduces MMAU, a new benchmark designed for audio large language models, akin to the role of MMLU and MMMU for text and vision. The authors hired domain experts to curate a challenging set of 10,000 human-annotated audio-question-response pairs, encompassing 27 distinct skills across unique tasks spanning speech, music, and environmental sounds. The authors further benchmarked 18 open-source and proprietary large audio-language models, highlighting the challenges in auditory understanding faced by current models.

All reviewers found the paper solid and recommended acceptance, and I agree with the reviewers that having a unified benchmark will be helpful for standardizing evaluation of large audio models.

**Additional Comments On Reviewer Discussion:**

The authors provided detailed and immediate updates to the paper during the discussion period which seemed to address reviewer concerns, so all reviewers recommended acceptance.

---

### Decision · Program_Chairs · 2025-01-22

Accept (Spotlight)